# A Systematic Review of Pneumonitis Following Treatment with Immune Checkpoint Inhibitors and Radiotherapy

**DOI:** 10.3390/biomedicines13040946

**Published:** 2025-04-12

**Authors:** Melina Yerolatsite, Nanteznta Torounidou, Anna-Lea Amylidi, Iro-Chrisavgi Rapti, George Zarkavelis, Eleftherios Kampletsas, Paraskevi V. Voulgari

**Affiliations:** 1Department of Medical Oncology, University Hospital of Ioannina, 45500 Ioannina, Greece; m.yerolatsite@gmail.com (M.Y.); nadia.torou@gmail.com (N.T.); annalea.ami@gmail.com (A.-L.A.); gzarkavelis@outlook.com (G.Z.); lekable4@yahoo.gr (E.K.); 2Department of Rheumatology, School of Health Sciences, Faculty of Medicine, University of Ioannina, 45110 Ioannina, Greece; iro.rapti@gmail.com

**Keywords:** pneumonitis, immune checkpoint inhibitors, radiotherapy

## Abstract

**Background:** Immune checkpoint inhibitors (ICIs) are increasingly included in management guidelines for various types of cancer. However, immune-related adverse events (irAEs) are an inevitable consequence of these therapies. Some of these side effects, such as pneumonitis, can be particularly serious. Additionally, the combination of ICIs with radiotherapy (RT) may further increase the risk of pneumonitis. **Objective:** The aim of this systematic review is to examine all available studies on pneumonitis following the use of ICIs and RT to assess its appearance and severity. **Methods:** We systematically searched four different databases (PubMed, Scopus, Cochrane, and DOAJ) to identify all relevant studies within our scope. Additionally, we reviewed the references of the studies we found, as well as those of other systematic reviews and meta-analyses. We assessed the risk of bias using the Cochrane Risk of Bias Tool version 2 for randomized controlled trials and the RTI Risk of Bias Item Bank for non-randomized trials. Finally, we extracted relevant data into an Excel file and presented them in tables throughout this study. **Results:** A total of 58 articles met our inclusion criteria, comprising 4889 patients across multiple studies and nine case reports. Due to significant heterogeneity in study methodologies and data reporting, a cumulative statistical analysis was not performed. The included studies were published between 2017 and 2025. The incidence of pneumonitis varied, with retrospective studies showing higher rates compared to randomized and non-randomized controlled trials. Case reports described a range of pneumonitis presentations, treatments, and outcomes, with corticosteroids being the primary treatment. **Conclusions:** The incidence of pneumonitis varied, with retrospective studies showing the highest rates compared to other study designs. Early detection and management of pneumonitis in patients receiving RT and/or ICIs are crucial for improving outcomes. Identifying high-risk patients through predictive models, radiomics, and biomarkers may help tailor treatment strategies and minimize toxicity. Further research is needed to establish the most appropriate diagnostic criteria, optimize management approaches, and refine advanced imaging and biomarker-based risk stratification to improve patient care. Interdisciplinary collaboration is essential for reducing the risk of pneumonitis and improving patient outcomes.

## 1. Introduction

Immune Checkpoint Inhibitors (ICIs) play an important role in the management of various types of cancer and are continuously being incorporated into worldwide guidelines. Moreover, radiotherapy (RT) remains a major treatment option for many patients, both as a radical therapy and as palliative care [1]. ICIs target signaling pathways involving cytotoxic T-lymphocyte antigen-4 (CTLA-4) and programmed death-1/programmed death-ligand 1 (PD-1/PD-L1), blocking tumor immune evasion and enhancing the immune response [1]. ICIs are associated with immune-related adverse events (irAEs), affecting multiple organs, including the skin, gastrointestinal tract, endocrine organs, and lungs. These toxicities can arise early in treatment or even months after discontinuation, sometimes requiring immunosuppressive intervention. The mechanisms behind irAEs remain unclear but may involve disrupted self-tolerance, immune cross-reactivity between tumor and normal tissues, inflammatory mediators, off-target effects, and microbiome influences. Genetic and environmental factors likely contribute to irAE development as well [2].

Not only do ICIs have some side effects, but RT has as well. Side effects resulting from the disruption of immune checkpoint signaling by ICIs are known as immune-related adverse events (irAEs). These can affect any organ system and vary in onset and severity. Notably, irAEs are common, occurring in approximately 80% of patients receiving combination therapy. Therefore, physicians must remain vigilant in monitoring and managing these irAEs [3,4].

RT also has side effects, which depend on the site of radiation. Generally, fatigue is the most common side effect of RT, regardless of the radiation site. In the case of thoracic RT, the most serious side effect is radiation-induced pneumonitis. This can occur during RT for lung cancer, esophageal cancer, breast cancer, and other types of cancer requiring thoracic RT [5].

The classification of pneumonitis consists of five grades (according to the Common Terminology Criteria for Adverse Events (CTCAE)). Grade 1 is asymptomatic, requiring only clinical or diagnostic observation without the need for further intervention. Grade 2 presents with mild symptoms that limit activities of daily living and require medical intervention. Grade 3 involves more severe symptoms that restrict self-care activities, necessitating medical intervention and oxygen therapy. Grade 4 is a life-threatening condition that requires urgent intervention. Finally, Grade 5 refers to fatal pneumonitis resulting in death [6].

The management of both lung and esophageal cancer has evolved over the years, with immunotherapy playing an increasingly important role in both types. Recently, the PACIFIC trial demonstrated that the addition of durvalumab to chemoradiotherapy in unresectable Stage III Non-Small Cell Lung Cancer (NSCLC) improved both progression-free survival (PFS) and overall survival (OS). An estimated 42.9% of patients randomly assigned to durvalumab survive for five years, while 33.1% remain alive and free of disease progression [7]. As a result, the incorporation of durvalumab into the management of unresectable locally advanced lung cancer has been included in clinical guidelines [8].

Moreover, the use of immunotherapy in the metastatic setting is well established, with multiple ICIs having been used for many years. Additionally, the role of immunotherapy in both limited-stage and extensive-stage small cell lung cancer (SCLC) is also well established [8]. Specifically, the ADRIATIC trial demonstrated an improvement in OS and PFS with the addition of durvalumab as consolidation therapy in limited-stage SCLC [9,10].

According to CheckMate 577, the addition of adjuvant nivolumab in patients with resectable esophageal cancer who received neoadjuvant chemoradiation therapy and had residual pathological disease results in longer disease-free survival [9]. Moreover, in metastatic esophageal cancer, RT may be considered a palliative therapy, and, in this setting, immunotherapy plays an important role in the first line of treatment [10].

Although both ICIs and RT in the thorax can provoke pneumonitis, which is a dangerous complication, there is debate about whether combining ICIs with RT leads to pneumonitis more frequently than using either therapy alone [10,11].

Three models have been created for the prediction of ≥grade 2 radiation pneumonitis. The first is the Quantitative Analyses of Normal Tissue Effects in the Clinic (QUANTEC) radiation pneumonitis model, which considers mean lung dose (MLD) as the sole predictive factor. As a landmark model, QUANTEC synthesizes data from over 70 published studies, making it the most widely used radiation pneumonitis model and the foundation for subsequent models. The second model analyzed was the Appelt model, which expands upon QUANTEC by incorporating six additional patient-specific variables: age, chemotherapy history, preexisting pulmonary comorbidities, smoking status, and tumor location. Lastly, the recently developed Thor model was evaluated. This model builds upon both QUANTEC and Appelt but simplifies the approach by including only two patient-specific variables—preexisting pulmonary comorbidities and age [11].

In the KEYNOTE-001 Phase I and Phase III PACIFIC trials, the treatment sequence was reversed, with patients receiving RT followed by immunotherapy. In the KEYNOTE-001 trial, a secondary analysis of 97 patients revealed that radiation pneumonitis developed in 15 out of 24 patients (63%) who had previously undergone chest RT. Additionally, pneumonitis recurred after sequential immunotherapy in three patients (13%) with prior chest RT. The overall incidence of pneumonitis across all grades was significantly higher in patients receiving combination therapy compared to those who did not (13% vs. 1%, *p* = 0.046) [12,13].

The Phase III PACIFIC trial evaluated the safety and toxicity of immunotherapy in 713 patients treated within two months after RT. The findings showed that the incidence of full-scale pneumonitis was notably higher in the combination therapy group compared to the RT-only group (33.9% vs. 24.8%) [13,14].

Additionally, an important topic for discussion is whether pneumonitis resulting from the combination of both treatments is more frequent or more severe than when it occurs with either treatment alone. For this reason, we conducted a systematic review of the literature to gather all available studies on this topic.

## 2. Methods

This systematic review is reported in accordance with the Preferred Reporting Items for Systematic Reviews and Meta-Analyses (PRISMA) statement [15]. It is also registered in the PROSPERO database (Registration code: PROSPERO 2025 CRD420251004671).

### 2.1. Search Strategy

Our search strategy was deliberately broad to ensure comprehensiveness and include all possible studies reporting the incidence of pneumonitis due to both RT and immunotherapy, allowing us to present their characteristics. For this reason, we used the following search algorithm: pneumonitis AND (immunotherapy OR immune checkpoint inhibitors OR ICIs) AND (radiotherapy OR RT) AND cancer. Furthermore, we applied this algorithm to four different databases (PubMed, Cochrane, Scopus, and DOAJ) to ensure a comprehensive retrieval of literature relevant to our study. Additionally, we reviewed the references of all included studies and relevant meta-analyses to ensure the inclusion of all relevant studies.

### 2.2. Study Inclusion and Exclusion Criteria

We included all articles relevant to our subject, regardless of whether they were case reports or studies involving multiple patients. All articles published before the search date (29 January 25) were included. Additionally, studies that provided mixed information on combination therapy or focused exclusively on one type of therapy—making it difficult to extract relevant data—were excluded. Furthermore, we excluded studies that did not specifically address the clinical question, as they had limited applicability to the research objective. Finally, studies where the full text was not available or was published in a language other than English or French were excluded.

### 2.3. Article Selection and Data Extraction

The selection of eligible articles and data recording were conducted by two different authors (MY and NT), and the data were compared at the end of the extraction process. Initially, we evaluated only the titles and abstracts, discarding articles that did not meet our criteria. Subsequently, we thoroughly reviewed the remaining articles to determine their eligibility for our study. In the case of disagreement, consultation with a senior author (GZ) and team consensus followed. Finally, duplicate articles were removed. A flowchart of the process was created (Figure 1). We then assessed the risk of bias in the relevant studies using the RTI Risk of Bias Item Bank.

The extracted data were first reviewed for accuracy and then entered into an electronic database. This database recorded various details, including the paper title, author, year of publication, patient gender and median age, sample size, type of cancer, histological type, cancer stage, smoking history, presence of preexisting pulmonary disease, epidermal growth factor receptor tyrosine kinase (EGFR) or anaplastic lymphoma kinase (ALK) status, PD-L1 expression, whether RT or immunotherapy was administered first, whether chemotherapy was given, type of chemotherapy, type of immunotherapy, type of RT, RT dose, RT site, number of patients who developed pneumonitis, pneumonitis grade, whether pneumonitis occurred inside or outside the radiation area, symptoms experienced during pneumonitis, CT imaging findings, pneumonitis outcomes, overall survival of patients with or without ≥grade 2 pneumonitis, risk of mortality, and control patient data from each study.

Two reviewers assessed independently the methodological quality of all the included studies. The risk of bias in the included randomized controlled trials was assessed by using the Cochrane Risk of Bias Tool for randomized trials version 2* [16]. The Research Triangle Institute (RTI) item bank* was used for the assessment of risk of bias in the non-randomized studies. Discrepancies were resolved by consensus [17]. Appendix A show the assessment of all included studies.

### 2.4. Data Synthesis

Taking into consideration the significant amount of heterogeneity that was present in the study designs, the methodology, the population, and the interventions among the included studies, we chose not to perform a quantitative synthesis of the study results. Thus, a narrative synthesis of the findings was followed. Table 1 and Table 2 show a summary of the data search and the search algorithm.

## 3. Results

A total of 58 articles met our inclusion criteria. Of these, 49 were studies involving multiple patients [11,14,18,19,20,21,22,23,24,25,26,27,28,29,30,31,32,33,34,35,36,37,38,39,40,41,42,43,44,45,46,47,48,49,50,51,52,53,54,55,56,57,58,59,60,61,62,63,64,65,66,67] and 9 were case reports [62,63,64,65,66,68,69,70,71]. One case report described two different patients with pneumonitis. Due to the significant heterogeneity in study methodologies and the variation in recorded data, we did not perform a cumulative statistical analysis of the pooled evidence. Moreover, most studies did not include data on control patients. However, we decided to include all available data on this subject rather than exclude articles without a control group.

The articles were published between 2017 and 2025, with a total of 4889 patients. Only 1 article was published in 2017, 1 in 2019, 13 in 2020, 9 in 2021, 9 in 2022, 7 in 2023, 8 in 2024, and 1 in 2025. Regarding the case reports, we report on nine patients. These case reports were published earlier than the larger studies. Specifically, the first case report was published in 2017. One was published in 2019, two others were published in 2020, followed by two in 2022. Only one was published in 2023, and two were published in 2024.

To evaluate the data from these articles, we separated them into evidence from retrospective studies with real-world data and evidence from randomized clinical trials. It was important to assess whether there were differences [11,14,18,19,20,21,22,23,24,25,26,27,28,29,30,31,32,33,34,35,36,37,38,39,40,41,42,43,44,45,46,47,48,49,50,51,52,53,54,55,56,57,58,59,60,61,62,63,64,65,66,67] between the results of the clinical trials and the retrospective studies. Moreover, we separately examined the case reports. Finally, we categorized the results based on whether patients received RT or ICIs first.

### 3.1. Radiation Pneumonitis and Immune-Related Pneumonitis

Through the studies, radiation pneumonitis and immune-related pneumonitis have been defined. The characteristic symptoms of both types of pneumonitis are nonspecific. It is considered a clinical diagnosis of exclusion, based on symptoms, timing of RT, imaging findings, and ruling out other causes of pneumonitis. Imaging of RT-induced pneumonitis typically shows involvement in the area of the lung that was irradiated [17].

Due to the morphological diversity of pneumonitis in patients receiving RT and ICIs, diagnosing pneumonitis based on traditional clinical characteristics and CT morphological features is often imprecise and subjective. Radiomics-based feature analysis offers a novel approach to predicting and differentiating diseases. Radiomic textures encompass a wealth of information, including subtle variations in shape, intensity, gradient, and texture, which are not discernible to the naked eye. These high-dimensional features extracted from medical images reveal the spatial heterogeneity of diseases. Furthermore, radiomics’ non-invasive nature provides an advantage over immunohistochemical analysis, which requires tissue samples obtained through surgery or biopsy. In a study by Colen RR. et al., the authors identified the most predictive radiomic features of immune-related pneumonitis, including skewness (a measure of histogram symmetry) and angular variance of the sum of squares (which quantifies the dispersion of voxel intensity distribution) [72,73]. Similarly, Wei Jiang and colleagues demonstrated that radiation pneumonitis was significantly associated with 11 radiomic features [74].

Generally, there are many confounding factors, such as a larger irradiated volume and potentially frailer patients with heart failure or disease progression, that may be related to RT-induced pneumonitis. However, these factors remain important considerations when assessing quality of life and determining which patients would benefit from palliative treatment [1].

In Hindocha’s study, it was reported that the incidence of pneumonitis in patients treated with a combination of ipilimumab and nivolumab was 3.75%, which was lower than the incidence observed with single-agent ICI therapy [1]. In addition, Geng’s meta-analysis reports that ICI-related pneumonitis most commonly occurs in patients receiving combination treatment [75]. A higher incidence of ICI-induced pneumonitis has been noted in patients with non-small cell lung cancer and squamous cancers [1].

Moreover, a higher incidence of pneumonitis has been observed in patients treated with PD-1 inhibitors compared to those treated with PD-L1 inhibitors. A potential mechanism behind this finding is that PD-1 inhibitors block the binding of PD-1 to PD-L2, which, in turn, enhances PD-L2 binding to the receptor-repulsive guidance molecule B. This leads to local T-cell clonal expansion, disrupting the balance of respiratory tolerance and thereby increasing the risk of pneumonitis [1].

### 3.2. Retrospective Studies: RT Before ICIs

This category includes 29 studies with a total of 2946 patients [11,18,19,20,21,22,23,24,25,26,27,28,29,30,31,32,33,34,35,36,37,38,39,40,41,42,43,44,45,46,47,48]. In all cases except one (Saade LJ), the majority of patients were male. In most patients, pneumonitis is categorized as Grade 1–2 or Grade 3–5. For this reason, we present the data from those studies in the same manner. Clinically significant pneumonitis is defined as Grade 3 or higher. Therefore, this categorization is also clinically important. In the Saade LJ study, the male-to-female ratio was equal. Additionally, only the study by Lv X included patients diagnosed with esophageal cancer, making it the only trial in which the patient population had esophageal cancer rather than lung cancer.

Moreover, four studies contained mixed data regarding whether RT was administered before or after ICIs. In eight studies, the sequencing of RT and ICIs was not explicitly reported; however, based on established guidelines, these studies were still included in this section [11,18,19,20,21,22,23,24,25,26,27,28,29,30,31,32,33,34,35,36,37,38,39,40,41,42,43,44,45,46,47,48].

Histologically, most studies reported adenocarcinoma as the predominant cancer type, followed by squamous cell carcinoma, with some studies including only patients with SCLC. The majority of patients were former or active smokers, except for Jung H.A.’s study, where nonsmokers outnumbered smokers [14]. Regarding disease stage, Stage III was the most common, followed by Stage IV, which aligns with approved treatment guidelines [11,18,19,20,21,22,23,24,25,26,27,28,29,30,31,32,33,34,35,36,37,38,39,40,41,42,43,44,45,46,47,48].

In all studies where data were available, patients received chemotherapy. No specific trend was observed in the choice of ICIs across studies, though PD-1 and PD-L1 inhibitors were the most commonly used. Durvalumab was the most frequently reported ICI. RT dose information was not included in most studies; however, in those that reported it, the dose was typically above 50 Gy [11,18,19,20,21,22,23,24,25,26,27,28,29,30,31,32,33,34,35,36,37,38,39,40,41,42,43,44,45,46,47,48].

Pneumonitis was assessed in all studies, with most cases classified as Grade I or II. The most commonly reported symptoms were dyspnea, cough, and fever. In eight studies, a small number of patients died either due to pneumonitis itself or disease progression resulting from pneumonitis-related treatment limitations [11,18,19,20,21,22,23,24,25,26,27,28,29,30,31,32,33,34,35,36,37,38,39,40,41,42,43,44,45,46,47,48].

Seventeen studies reported that patients died due to pneumonitis, either as a direct result of its symptoms or due to disease progression following a lack of improvement after its onset. In Shaverdian N.’s study, one patient died; in Landman Y.’s study, one patient died; in Sugimoto T.’s study, one patient died; in Lv X.’s study, one patient died; and in Yamanaka Y.’s study, one patient died due to pneumonitis. Additionally, in Altan M.’s study, five patients died due to pneumonitis, and, in Preti BTB’s study, three patients died from pneumonitis [11,19,22,28,29,31,34].

All the characteristics of the included articles are summarized in Table 3 and Table 4.

### 3.3. Retrospectives Studies: ICIs Before RT

Only nine studies examined pneumonitis after RT and ICIs, with ICIs administered first [47,48,49,50,51,52,53,54,76]. Table 5 summarizes all the characteristics of these studies, which included a total of 1041 patients. In this section, the majority of patients were male. Additionally, in all studies except one, the majority of patients were former or active smokers. In Gao Y’s study, approximately half of the patients were nonsmokers and half were smokers.

Regarding histology types, squamous cell carcinoma and adenocarcinoma were the most common across all studies. Similarly, Stage III and Stage IV were the most frequently reported disease stages. In all studies except one, PD-1 inhibitors were the most commonly used immunotherapy, followed by PD-L1 inhibitors. Furthermore, in the studies that reported RT doses, the dose was greater than 50 Gy [47,48,49,50,51,52,53,54,67].

Pneumonitis was observed in approximately 50% of patients in each study, except for Gao Y’s study, where the incidence was lower at approximately 15%. In most studies, pneumonitis severity was evenly distributed among Grades 1, 2, and 3–5. For this reason, we also maintain this categorization in our review. Only two studies provided information on the timing of pneumonitis onset in relation to the initiation of immunotherapy. In Saito S’s study, the median months after the initiation of ICIs were 3 months for grade 3 pneumonitis and 5 months for grade 2 [49]. In addition, in Lu X’s study, the median months were 4.3 months [50]. Finally, the treatment for pneumonitis was the use of steroids in all the studies that reported this specific information [47,48,49,50,51,52,53,54,67].

All the characteristics of the included articles are summarized in Table 5.

### 3.4. Clinical Trials

Four randomized controlled trials were found, with a total of 1603 patients [9,14,54,58]. All participants in these trials were diagnosed with lung cancer and were predominantly male (Table 6). All patients also received radiotherapy (RT) before therapy with immune checkpoint inhibitors (ICIs).

The majority of clinical trials focused on ≥Grade 2 pneumonitis, and, in each study, the incidence of pneumonitis was approximately 15–25%. Additionally, the randomized evidence shows that most cases of pneumonitis (75%) occurred within the first 12 weeks, aligning with the peak risk period for radiation pneumonitis. Only one patient developed pneumonitis after 18 weeks, suggesting a significant decline in risk beyond the 12-week window [14,54,56].

Finally, two studies reported that patients died either due to pneumonitis itself or due to disease progression resulting from a lack of improvement in pneumonitis symptoms, which prevented the initiation of further therapy. In Peters S.’s study [31], two patients died. In Durm G.A.’s study, one patient died due to pneumonitis [55].

In addition, there are six non-randomized clinical trials with a total of 384 patients [55,56,57,59,60,61]. The majority of patients in these trials were male and smokers. Moreover, all patients had lung cancer and received RT before ICIs, as well as chemotherapy. Most of these patients were at Stage 3 or Stage 4 or had extensive small cell lung cancer (SCLC). Pneumonitis occurred in approximately 15–20% of patients in each study. However, in Welsh J.W.’s study, none of the patients experienced pneumonitis, making it the only study with no reported cases of pneumonitis [60].

All the characteristics of the included articles are summarized in Table 6. Table 7, Table 8 and Table 9 include all the relevant data on pneumonitis for all patients.

### 3.5. Case Reports

There are nine different studies referring to case reports of patients who suffered from pneumonitis due to RT and ICIs [63,64,65,66,68,69,70,71,77]. One of these studies included two separate case reports [59]. Six of the patients were male, while the gender of the remaining four was not specified. The median age of the patients was 68 years. Three patients were active or former smokers, four were non-smokers, and smoking status was not specified for the remaining three.

Additionally, six patients had lung cancer, two had melanoma, one had esophageal cancer, and one had colon cancer. The median RT dose was 50.4 Gy.

The symptoms of pneumonitis included cough, dyspnea, fever, and fatigue. Computed tomography imaging revealed patchy consolidation within the RT field, and, in some cases, outside of this field. In two cases, extensive acute interstitial pneumonia was observed. Furthermore, fibrosis developed in one patient. One patient remained asymptomatic from the diagnosis of pneumonitis until the complete resolution of the disease [63,64,65,66,68,69,70,71,77].

The treatment for pneumonitis initially consisted of intravenous corticosteroids. Once the patient’s clinical condition improved, the treatment was transitioned to oral corticosteroids. All patients, except two, showed improvement with corticosteroid therapy, and no recurrences of pneumonitis were observed. However, in two patients, pneumonitis recurred during corticosteroid tapering. In one of these cases, an increased dose of corticosteroids was required [63,64,65,66,68,69,70,71,77].

Table 10 summarizes all the case reports.

## 4. Discussion

A total of 58 articles were included in this study, comprising 49 studies with multiple patients and 9 case reports. Given the substantial heterogeneity in study methodologies and data reporting, a cumulative statistical analysis was not feasible. Retrospective studies reported higher pneumonitis rates compared to randomized clinical trials [11,14,18,19,20,21,22,23,24,25,26,27,28,29,30,31,32,33,34,35,36,37,38,39,40,41,42,43,44,45,46,47,48,49,50,51,52,53,54,55,56,57,58,59,60,61,62]. Case reports described diverse pneumonitis presentations, management strategies, and outcomes, with corticosteroids being the primary treatment [63,64,65,66,68,69,70,71,77].

Diagnosing pneumonitis remains challenging and is primarily based on excluding infections through clinical assessment, imaging, oxygen saturation, pulmonary function tests, and blood cultures. Differentiating radiation-induced lung injury (RILI) from immune-related lung injury (IRLI) is particularly difficult. Treatment for severe cases (Grade 3/4) involves systemic corticosteroids and avoidance of the underlying cause. If no improvement is seen within 48 h, consensus guidelines should be followed. Biomarkers, cytokine levels, and lymphocytic profiles are being explored for better diagnosis. Thorough pre-treatment evaluation, including pulmonary and cardiac function tests, imaging, and bronchoalveolar lavage (if needed), is essential. Advanced imaging (computer tomography (CT), positron emission tomography/computed tomography (PET/CT)) helps distinguish other conditions. Prior treatments, patient compliance, and concurrent therapies must be carefully assessed to minimize toxicity risks, with all complex cases requiring interdisciplinary discussion [67].

Both RT and immunotherapy can lead to lung injury by triggering an excessive release of cytokines, which cause inflammation and tissue damage through direct and indirect mechanisms. Directly, cytokines such as transforming growth factor beta (TGF-β) and Tumor Necrosis Factor Alpha (TNF-α) activate pathways like TGF-β/Smad and TNF-α/nuclear factor-kappa B (NF-κB), leading to fibrosis and inflammation. Indirectly, cytokines such as interleukin (IL)-4, IL-6, IL-10, and IL-17 recruit immune cells like neutrophils, macrophages, and lymphocytes, which further amplify lung injury. Additionally, oxidative stress (reactive oxygen species/reactive nitrogen species (ROS/RNS) pathway) and immune activation (cyclic GMP-AMP synthase/stimulator of interferon genes (cGAS-STING) pathway) contribute to early lung damage. The overlapping roles of cytokines, particularly IL-3, IL-6, IL-10, IL-17, TNF-α, and TGF-β, suggest significant crosstalk between pathways in radiation-induced lung injury (RILI) and immunotherapy-related lung injury (IRLI). Given TGF-β’s dual role in promoting lung fibrosis and influencing the tumor microenvironment, inhibiting this pathway is considered a promising strategy for reducing lung toxicity while enhancing the effectiveness of immunotherapy [78,79].

Specifically, Schoenfeld’s study reported a case of a patient with advanced melanoma who developed pneumonitis following radiotherapy combined with the PD-1 inhibitor nivolumab. The study found that elevated levels of C-X-C chemokine receptor type 2 (CXCR2), IL-1 receptor antagonists, and IL-2 receptor antagonists were associated with the onset and progression of pneumonitis, suggesting a potential link between these immune markers and treatment-induced lung inflammation [70]. However, none of the included articles examine these pathways or the role of these cytokines in the development of pneumonitis [11,14,18,19,20,21,22,23,24,25,26,27,28,29,30,31,32,33,34,35,36,37,38,39,40,41,42,43,44,45,46,47,48,49,50,51,52,53,54,55,56,57,58,59,60,61,62,63,64,65,66,68,69,70,71,77].

CD73 and adenosine are key regulators of lung homeostasis and inflammation, contributing to an immunosuppressive tumor microenvironment. The CD73/adenosine pathway can hinder the effectiveness of cytotoxic therapies by degrading ATP, a pro-inflammatory molecule released during therapy-induced cell damage. Targeting CD73 may improve the effectiveness of radio-immunotherapy by reducing tumor immune escape, but it also raises concerns about increased immune-related adverse events (irAEs) in normal tissues. Combined radio-immunotherapies already carry a higher risk of irAEs in the lungs, and further inhibition of CD73 could exacerbate pulmonary toxicity, limiting therapeutic outcomes [80].

In Saito’s study, various factors were examined to determine the cause of pneumonitis. Patient factors, including age, sex, ICI agent, interval between RT and ICI, and prior history of ICI before RT, were not significantly associated with grade ≥2 pneumonitis. However, the presence of obstructive pneumonia at the time of RT was statistically significant (*p* = 0.036). The explanation for this is that obstructive pneumonia reduces pulmonary reserve, leading to more severe pneumonitis compared to patients with normal lung function. Other factors, such as tumor volume, Brinkman index, dosimetric parameters (lung V5, V10, V20, V30, and MLD), lactate dehydrogenase, and C-reactive protein, did not show significant differences between patients with grade ≤1 pneumonitis and those with grade ≥2 pneumonitis [49].

Moreover, Stana’s study identifies the diffusing capacity for carbon monoxide (DLCO) as a predictor of pulmonary toxicity and a valuable tool for monitoring post-treatment lung function, reinforcing the importance of peri-treatment lung function testing in optimizing radiation planning. Future prospective studies with larger cohorts should incorporate a pre-treatment DLCO threshold of 60% as a criterion for patient selection to minimize pulmonary toxicity in those with reduced lung reserve. Additionally, V65–45% could serve as a complementary dosimetric parameter for predicting lung function outcomes after therapy [81].

In addition, a meta-analysis was performed, examining only some of the relevant studies. Balasubramanian A.’s meta-analysis [82] revealed higher pooled rates of G3 pneumonitis in other studies compared to PACIFIC (Antonia S.J.’s study) [14], which did not report G2 pneumonitis rates. Additionally, several studies included in the meta-analysis did not report G1 pneumonitis rates, as asymptomatic radiographic changes related to radiotherapy were considered clinically insignificant [14,82].

In Li Y’s meta-analysis, the difference in the rate of pneumonitis between real-world data and clinical trials was evaluated. Specifically, real-world studies reported an incidence of pneumonitis across all grades of 46.5% (95% CI, 31.1–62.3%), whereas clinical trials reported a lower incidence of 31.3% (95% CI, 22.8–39.8%) [83]. This discrepancy may be due to differences in patient populations, as many patients in real-world settings would not meet the eligibility criteria for clinical trials. Therefore, it is crucial to assess the findings of observational real-world studies while acknowledging their limitations [79,80]. Although, due to heterogeneity, we cannot perform a quantitative analysis of our data, there appears to be a trend suggesting that pneumonitis is more common in retrospective trials compared to clinical trials. This may be due to the selection of patients included in clinical trials, as well as the more thorough follow-up between therapy appointments. This trend has also been observed in other systematic reviews and meta-analyses, although with fewer studies than in our review [75,82,84,85]. For instance, Kuang Y.’s study concluded that the incidence of severe and fatal radiation pneumonitis in patients with unresectable stage III NSCLC treated with concurrent chemoradiotherapy ranges from 3.62% to 7.85%, with a higher incidence reported in real-world data studies [86].

In addition, Yang Z.’s meta-analysis concluded that the timing of ICI initiation after RT is crucial. When the interval between the completion of radiotherapy and the initiation of durvalumab treatment was shorter than 42 days, the incidence of pneumonitis (Grade ≥ 3) was 4.12% [95% CI (0.02, 0.06), I^2^ = 0.00%, *p* = 0.56] [87].

Post-radiotherapy and oncological follow-up are essential to monitor and manage late adverse effects through regular clinical assessments, lab tests, and imaging. Lung tissue damage, particularly pneumonitis, can progress to pulmonary fibrosis, impairing lung function. The risk of fibrosis depends on patient-, treatment-, and tumor-related factors, with some pretreatment risks potentially leading to lung injury even after successful therapy. Considering these factors, advanced treatment approaches like radioimmunotherapy and radiochemoimmunotherapy offer promising, safe, and effective options for patients with locally advanced NSCLC [88].

The factors that may be responsible for the appearance of pneumonitis were also examined in studies in which the patients received either immunotherapy or RT. For instance, Cui P.’s study states that immune-mediated pneumonitis is likely to preferentially be presented in patients with characteristics that worsen pulmonary conditions, including smoking status, prior treatment, combination therapy, primary tumor type, and prior lung disease [89]. In Yamaguchi T.’s study, pre-existing interstitial lung disease is associated with an increased risk of drug-induced pneumonitis [90]. Rahi M.S.’s study discusses the causes of radiation-induced pneumonitis, emphasizing that it is influenced by both treatment-related and patient-related risk factors. Higher total radiation doses (>40 Gy), larger irradiated lung volumes, and fractionation patterns significantly increase the risk, while advanced irradiation techniques like IMRT, SBRT, and proton therapy help reduce it. Concurrent chemotherapy and immune checkpoint inhibitors (ICIs) further elevate pneumonitis risk, though data on ICIs remain mixed. Among patient-related factors, older age (>70 years) is associated with a higher risk due to reduced cardiopulmonary reserve. The role of smoking and COPD remains controversial, with some studies suggesting a protective effect of smoking. However, pre-existing interstitial lung disease (ILD) is a well-established risk factor for severe pneumonitis and increased mortality. Tumor characteristics, such as mid-lower lung zone involvement, larger tumor volume, and concurrent endocrine therapy in breast cancer patients, also contribute to a higher risk. While newer technologies and predictive models (e.g., 4D CT, FDG imaging, NTCP models) aim to minimize risk, further research is needed to optimize treatment strategies and reduce toxicity [91].

In addition, in most studies, including both case reports and larger studies, the treatment of pneumonitis involved the use of corticosteroids in varying doses. Early diagnosis and prompt initiation of corticosteroid treatment were the most effective strategies for managing pneumonitis [11,19,20,21,22,28,31,32,35,36,38,40,47,49,55,63,64,65,66,68,69,70,71,77].

This study has limitations, including the lack of a qualitative synthesis of the results. Moreover, the high heterogeneity, due to differences in population, interpretations, study types, and methodological assessments, made it challenging to draw definitive conclusions. However, we chose to include all relevant studies in this systematic review rather than excluding most of them to minimize heterogeneity. Another limitation is the varying interpretations of pneumonitis across studies. Although it was not feasible to separate all the different definitions, we categorized them into two main groups and included all relevant articles in each.

In the future, it is important to conduct higher-quality studies, such as those focusing on specific populations or employing better-designed methodologies with larger sample sizes. This will provide more accurate data on this irAE, enabling earlier management to prevent it from becoming detrimental to patients’ quality of life or even life-threatening. Finally, real-world data are crucial for investigating the effects of different therapies on patients that oncologists encounter in clinical settings. For this reason, it is important to conduct more well-organized real-world studies to assess the effectiveness of therapies in real-time.

## 5. Conclusions

The incidence of pneumonitis following the use of RT and ICIs varied across studies, with retrospective studies showing higher rates compared to randomized and non-randomized controlled trials. Case reports highlighted the diverse presentations, treatments, and outcomes of pneumonitis, with corticosteroids being the primary treatment approach. Early detection and management of pneumonitis in patients receiving RT and/or ICIs are crucial for improving patient outcomes. Identifying high-risk patients through predictive models, radiomics, and biomarkers may help tailor treatment strategies and minimize toxicity. Collaboration with other specialists can significantly improve patient management and outcomes.

## Figures and Tables

**Figure 1 biomedicines-13-00946-f001:**
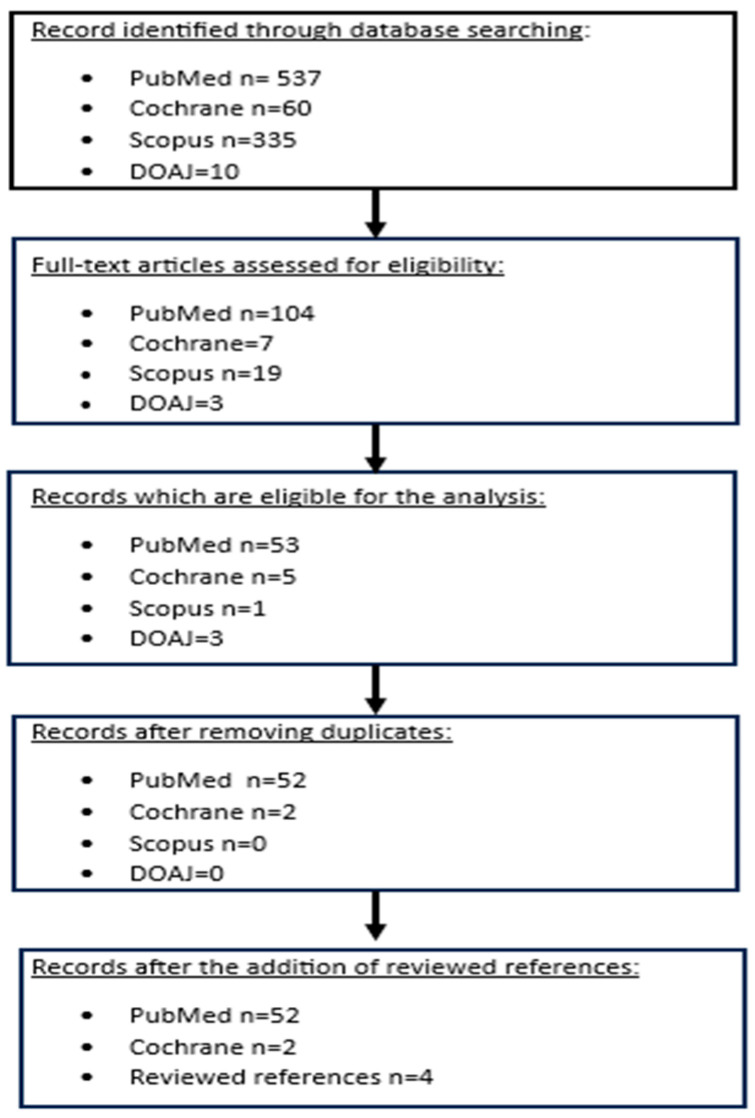
Flowchart depicting the search strategy followed in our analysis.

**Table 1 biomedicines-13-00946-t001:** Search strategy and boolean commands.

Search Setting	Details
Databases Searched	PubMed, Cochrane, Scopus, DOAJ
Search Date	2029/1/25
Study Period	All studies published before the search date
Languages Included	English, French
Study Types	Case reports, retrospective and prospective studies, clinical trials
Exclusion Criteria	Studies without full text, non-English/French articles, unrelated studies
Screening Process	Titles and abstracts reviewed, then full-text evaluation
Reviewer Process	Two independent reviewers (MY and NT), with GZ for conflict resolution

**Table 2 biomedicines-13-00946-t002:** Advanced search settings and boolean commands.

Database	Search Algorithm (Boolean Commands Used)
PubMed	(pneumonitis AND (immunotherapy OR “immune checkpoint inhibitors” OR ICIs) AND (radiotherapy OR RT) AND cancer)
Cochrane	(pneumonitis AND (ICIs OR “immune checkpoint inhibitors”) AND (RT OR radiotherapy) AND cancer)
Scopus	(TITLE-ABS-KEY (pneumonitis) AND TITLE-ABS-KEY (ICIs OR “immune checkpoint inhibitors”) AND TITLE-ABS-KEY (RT OR radiotherapy) AND TITLE-ABS-KEY (cancer))
DOAJ	(pneumonitis AND (ICIs OR “immune checkpoint inhibitors”) AND (radiotherapy OR RT) AND cancer)

**Table 3 biomedicines-13-00946-t003:** Characteristics of the included studies in which RT was administered first.

No	Author (Year)	Number of Patients	Gender (M/F)	Median Age	Cancer (Lung/Esophageal)	Histology (NonSquamous/Squamous/SCLC/Other)	Smoking (Yes/No/Unknown)	Relevant Chronic Diseases	Stage (I or II/III/IV)	RT Prior ICIs	Chemotherapy (Yes)	Type of ICIs (PD-1i, PD-L1i, Ctla4i)	Median Dose of RT (Gy)	Median Lung V20%
1	Shaverdian N (2020) [11]	62	33/29	66	62/0	NA	60/2/0	19	0/62/0	62	33	0/62/0	60	NA
2	Thomas HM (2022) [18]	21	10/11	64	21	11/8/0/2	20/1/0	7	0/18/3	21	21	0/21/0	NA	NA
3	Landman Y (2021) [19]	39	25/14	68	29	28/11/0/0	33/6/0	NA	0/39/0	39	NA	0/39/0	69.9	21
4	Jang JY (2021) [20]	51	11/40	62	51/0	29/22/0/0	37/14/0	9	5/46/0	51	51	0/51/0	66	22.6
5	Barron F (2020) [21]	40	NA	NA	40/0	NA	NA	NA	NA	40	NA	NA	NA	NA
6	Sugimoto T (2021) [22]	20	15/5	69.5	20/0	11/8/0/0	18/2	NA	0/20/0	20	20	0/20/0	NA	17.9
7	Cousin F (2021) [23]	80	45/35	69.5	80/0	30/37/0/13	76/4/0	80	NA	NA	75	0/11/0	66	17.7
8	Tjong M.C. (2022) [24]	65	35/29	70	64/0	56/8/0/0	48/15/0	NA	0/1/63	NA	57	24/40/0	NA	NA
9	Noda-Narita S (2022) [25]	257	NA	NA	257/0	NA	NA	NA	NA	NA	NA	NA	NA	NA
10	Saade LJ (2023) [26]	14	7/7	64.72	14/0	8/5/0/1	3/11	NA	0/14/0	14	14	0/14/0	NA	NA
11	Neibart SS (2023) [27]	550	300/250	NA	550	414/136/0/0	NA	220	0/NA/NA	NA	NA	NA	NA	NA
12	Atlan M (2023) [28]	140	74/66	67	140/0	79/55/0/0	121/19	46	0/140/0	140	140	0/140/0	66	NA
13	Preti BTB (2023) [29]	118	61/57	66.3	118/0	73/38/0/0	107/5/0	NA	47/55/52	118	118	0/118/0	NA	NA
14	Cai Z (2023) [3]	78	12/66	NA	78/0	0/0/78/0	52/26	NA	0/0/78	NA	78	32/44/0	NA	NA
15	Lv X (2024) [31]	97	66/31	64	0/97	0/97/0/0	50/47/0/0	22	56/63	NA	83	97/0/0	NA	21.3
16	Smesseim I (2024) [32]	45	27/18	64	45/0	22/20/0/3	41/3/0	23	0/45/0	45	45	0/45/0	NA	NA
17	Kraus KM (2024) [33]	39	NA	65.8	39/0	NA	NA	NA	NA	NA	13	NA	NA	NA
18	Yamanaka Y (2024) [34]	12	1/11	65.5	12/0	5/7/0/0	12/0	NA	0/12/0	NA	12	12/0/0	NA	NA
19	Murata S. (2024) [35]	100	77/23	66	100/0	55/31/0/0	85/15/0	70	0/100/0	100	100	0/100/0	NA	23.35
20	Jung H.A.(2020) [36]	21	19/2	65.9	21/0	6/15/0/0	5/16/0	NA	0/21/0	21	21	0/21/0	NA	NA
21	Chu CH (2020) [37]	31	26/5	64	31/0	20/8/0/3	23/8/0	NA	0/31/0	31/0	31	0/31/0	NA	NA
22	Miura Y (2020) [38]	41	8/33	72	41/0	21/6/0/0	33/8/0	NA	NA/35/NA	41	41	NA	NA	NA
23	Amino Y. (2020) [39]	20	18/2	72	20	6/14/0/0	19/1/0	NA	0/20/0/0	20	20	1/19/0	NA	NA
24	Chen D. (2020) [40]	33	16/24	NA	33/0	26/7/0/0	37/3/0	NA	0/0/33	33/0	NA	16/17/0	50	22
25	Inoue H. (2020) [41]	30	22/11	68	68/0	16/13/0/0	25/8/0	NA	0/0/30	30/0	30	0/30/0	NA	NA
26	Saad A. (2022) [42]	71	19/11	67	71/0	39/23/0/9	64/7/0	NA	0/0/71	71	71	0/71/0	61	NA
27	Wang C.C. (2021) [43]	61	12/49	63	61/0	42/15/0/4	43/18/0	NA	0/61/0	61	61	0/61/0	NA	NA
28	Desilets A. (2020) [44]	147	98/49	67	147/0	97/47/0/3	NA	NA	0/142/0	147	147	0/147/0	NA	NA
29	Tsukita Y. (2021) [45]	87	NA	NA	87/0	NA	NA	NA	0/87/0	87	87	0/87	NA	NA
30	Avrillon V. (2022) [46]	576	419/157	64	576/0	NA	NA	21	0/576/0	576	576	0/576	NA	NA

NA stands for “Not Available”.

**Table 4 biomedicines-13-00946-t004:** Characteristics of the included studies in which RT was administered first or second.

No	Author (Year)	Number of Patients	Gender (M/F)	Median Age	Cancer (Lung/Esophageal)	Histology (NonSquamous/Squamous/SCLC/Other)	Smoking (Yes/No/Unknown)	Relevant Chronic Diseases	Stage (I or II/III/IV)	ICIs Prior RT	Chemotherapy (Yes)	Type of ICIs (PD-1i,PD-L1i,Ctla4i)	Median Dose of RT (Gy)	Median Lung V20%
1	Chen Y (2021) [40]	96	73/23	60	96/0	33/32/0/28	96/0/0	7	0/14/82	96	NA	85/11/0	52	15.75
2	Bi J (2022) [54]	40	6/34	63	8/32	NA	26/14/0	2	NA	52	15	34/6/0	60	15.5
3	Bi J (2024) [47]	192	171/21	63	192/0	0/0/38/154	148/44/0	25	7/132/53/	192	52	171/21	60	20
4	Wang K (2024) [48]	118	100/18	62	118/0	57/59/0/2	65/53/0	23	4/46/68	118	NA	112/6/0	60	16.9

NA stands for “Not Available”.

**Table 5 biomedicines-13-00946-t005:** Characteristics of the included studies in which ICIs were administered first.

No	Author (Year)	Number of Patients	Gender (M/F)	Median Age	Cancer (Lung/Esophageal)	Histology (Squamous/Non Squamous/Unkown)	Smoking (Yes/No/Unknown)	Relevant Chronic Diseases	Stage (I or II/III/IV)	RT prior ICIs/ICIs prior RT	Chemotherapy (Yes)	Type of ICIs (PD-1i,PD-L1i,Ctla4i)	Median Dose of RT (Gy)	Median Lung V20%
1	Saito S (2021) [49]	29	24/5	68	29/0	NA	NA	11	0/0/29	17/12	NA	25/4/0	NA	NA
2	Lu X (2022) [50]	196	160/36	61	196/0	59/77/57/3	143/53	50	9/152/35	151/45	74	53/69	NA	19.1
3	Gao Y (2023) [51]	202	153/49	57.54	202/0	110/88/1/0	99/103/0	46	NA	160/42	NA	NA	54	NA
4	Yang Y (2023) [52]	123	106/17	63	192	50/73/0/0	93/30/0	NA	0/123/0	62/41	123	56/67/0	NA	NA
5	Song Z (2025) [53]	45	45/0	NA	45/0	NA	38/7/0	7	0/45/0	29/16	38	0/45/0	NA	NA

NA stands for “Not Available”.

**Table 6 biomedicines-13-00946-t006:** Characteristics of the included clinical trial.

No	Author (Year)	Number of Patients	Gender (M/F)	Median Age	Cancer (Lung/Esophageal)	Histology (NonSquamous/Squamous/SCLC/Other)	Smoking (Yes/No/Unknown)	Relevant Chronic Diseases	Stage (I or II/III/IV) or Unkown	RT prior ICIs/ICIs Prior RT	Chemotherapy (Yes)	Type of ICIs (PD-1i, PD-L1i, CTLA4i)	Median Dose of RT (Gy)	Median Lung V20%
1	Vansteenkiste JF (2024) [55]	473	319/139	NA	473/0	207/251/0/0	298/160/0	125	0/473/0	NA	473	0/473/0	NA	NA
2	Dum GA (2020) [56]	93	59/33	66	92/0	51/41/0/0	87/5/0	NA	0/92/0	92/0	92	92/0	NA	NA
3	Peters S (2019) [57]	80	NA	NA	80/0	NA	NA	NA	NA	NA	80	NA	NA	NA
4	Antonia SJ (2017) [14]	476	334/142	64	476/0	252/224/0/0	433/43/0	NA	0/476/0	476/0	476	476/0	NA	NA
5	Perez A.B. (2020) [58]	21	13/8	66	21/0	0/0/21/0	NA	NA	21 extensive SCLC	21/0	21	0/21/0	30	NA
6	Peters S. (2022) [59]	78	50/28	61.1	78/0	0/0/78/0	78/0/0	NA	78 limited SCLC	78/0	78	78/0/78	45	NA
7	Welsh J.W. (2020) [60]	33	20/13	62	33/0	0/0/30/3	NA	NA	33 extensive SCLC	33/0	33	33/0/0	45	19
8	Welsh J.W. (2020) [61]	40	16/24	64	40/0	0/0/36/4	37/3/0	NA	40 limited SCLC	40/0	40	40/0/0	45	NA
9	Garassino M. (2022) [62]	117	73/44	68	117/0	63/45/0/9	108/9/0	NA	0/116/0	117/0	117	0/117/0	NA	NA
10	Kelly R.J.(2021) [9]	532	449/83	62	0/532	376/155/0/1	NA	NA	0/530/2	0/532	532	532/0/0	NA	

NA stands for “Not Available”.

**Table 7 biomedicines-13-00946-t007:** Incidence of pneumonitis in retrospective studies.

No	Author	Pneumonitis	Grade 1	Grade 2	Grade 3–5	Out of Irradiation Pneumonitis	Intra Irradiation Pneumonitis	Median Months After the Completion of Radiotherapy	Median Months After the Start of ICIs
1	Shaverdian N (2020) [11]	11		10	1	NA	NA	3.3	1.6
2	Saito S (2021) [49]	17	10	4	3	3	4	NA	3 months for grade 3 pneumonitis and 5 months for grade 2
3	Landman Y (2021) [19]	6	NA	NA	NA	NA	NA	NA	2.2
4	Jang JY (2021) [20]	40	10	27	3	NA	NA	2.3	1.4
5	Barron F (2020) [21]	16	3	9	4	NA	NA	NA	4.5
6	Chen Y (2021) [40]	47	0	28	19	NA	NA	NA	NA
7	Bi J (2022) [54]	26	10	7	9	NA	NA	2.2	NA
8	Sugimoto T (2021) [22]	3	0	2	1	NA	NA	NA	NA
9	Cousin F (2021) [23]	15	10	3	2	NA	NA	20.5	5
10	Lu X (2022) [50]	108	58	42	8	NA	NA	NA	4.3
11	Thomas HM (2022) [18]	10	0	NA	NA	NA	NA	NA	NA
12	Noda-Narita S (2022) [25]	34	NA	NA	NA	NA	NA	NA	NA
13	Saade Li (2023) [26]	8	3	NA	NA	NA	NA	3.62	2.45
14	Neibart SS (2023) [27]	19	NA	NA	NA	NA	NA	NA	NA
15	Gao Y (2023) [51]	36	15	10	11	NA	NA	NA	NA
16	Altan M (2023) [28]	32	0	24	8	NA	NA	5.2	2.56
17	Preti BTB (2023) [29]	47	4	26	20	NA	NA	NA	NA
19	Cai Z (2023) [30]	18	NA	NA	2	NA	NA	NA	NA
20	Bi J (2024) [47]	111	24	28	59	NA	NA	2.4	NA
21	Yang Y (2023) [52]	90	NA	NA	9	NA	NA	NA	NA
22	Lv X (2024) [31]	45	20	20	5	NA	NA	NA	4
23	Smesseim I (2024) [32]	14	NA	NA	NA	2	2	NA	NA
24	Kraus KM (2024) [33]	19	NA	NA	NA	NA	NA	NA	NA
25	Yamanaka Y (2024) [34]	12	3	4	5	NA	NA	NA	NA
26	Wang K (2024) [48]	57	27	28	2	NA	NA	2.8	NA
27	Song Z (2025) [53]	NA	NA	5	1	NA	NA	NA	NA
28	Tjong MC (2022) [24]	6	NA	1	5	1	4	NA	NA
29	Murata S (2024) [35]	81	46	31	4	NA	NA	NA	3.75
30	Jung HA (2020) [36]	17	NA	NA	NA	NA	NA	NA	NA
31	Chu CH (2020) [37]	5	NA	NA	2	NA	NA	NA	NA
32	Miura Y (2020) [38]	25	13	11	1	NA	NA	NA	NA
33	Amino Y (2020) [39]	1	NA	NA	1	NA	NA	NA	NA
34	Chen D (2020) [40]	7	NA	NA	4	NA	NA	NA	NA
35	Innoue H (2020) [41]	22	8	14	0	NA	NA	2.2	1.4 (After final dose)
36	Saad A (2022) [42]	35	NA	NA	4	NA	NA	NA	NA
37	Wang CC (2021) [43]	11	NA	NA	3	NA	NA	NA	NA
38	Desilets A (2020) [44]	44	NA	NA	9	11	33		1.8
39	Tsukita Y (2021) [45]	77	45	24	7	NA	NA	NA	NA
40	Avrillon V (2022) [46]	NA	NA	NA	6	NA	NA	NA	NA

NA stands for “Not Available”.

**Table 8 biomedicines-13-00946-t008:** Incidence of pneumonitis in randomized clinical trials.

No	Author	Total Events in Treatment	Total Patient in Treatment	Total Events in Control	Total Patients in Control
1	Vansteenkiste J (2024) [55]	94	476	33	237
2	Peters S (2022) [59]	22	78	4	75
3	Antonia SJ (2017) [14]	62	476	18	237
4	Kelly R.J. (2021) [9]	71	532	15	260

**Table 9 biomedicines-13-00946-t009:** Incidence of pneumonitis in non-randomized clinical trials.

No	Author	Pneumonitis	Grade 1	Grade 2	Grade 3–5	Out of Irradiation Pneumonitis	Intra Irradiation Pneumonitis	Median Months After the Completion of Radiotherapy	Median Months After the Start of ICIs
1	Durm GA (2020) [56]	NA	NA	10	5	NA	NA	NA	2
2	Perez AB (2020) [58]	5	NA	NA	2	NA	NA	NA	NA
3	Welsh JW (2020) [60]	0	0	0	0	0	0	0	0
4	Welsh JW (2020) [61]	6	NA	NA	3	NA	NA	NA	NA
5	Gerassimo M (2022) [62]	22	2	17	3	NA	NA	NA	NA
6	Peters S. (2019) [57]	26	NA	NA	8	NA	NA	NA	NA

NA stands for “Not Available”.

**Table 10 biomedicines-13-00946-t010:** Characteristics of the included case reports.

No	Author (Year)	Gender/Age	Smoking	Cancer/Stage	Type of RT/Dose	Type of ICIs	Duration of ICIs (Months)	Symptoms	CT Imaging	Therapy	Outcome
1	Louvel G (2017) [63]	NA/70	No	Melanoma/IV	SBRT 25 Gy	Pembrolizuamb	5	None	Peripheral consolidation	None	Still asymptomatic
		NA/59	No	Colon cancer/IV	SBRT 45 Gy	Atezolizumab	3	Cough	Patchy central area of consolidation	Corticosteroids	NA
2	Chen Y (2020) [64]	M/64	No	NSCLC/III	RT 58 Gy	Camrelizumab	After 8th cycle	Fever, dyspnea and cough	Patchy consolidation and ground-glass opacities	Stop ICIs/Corticosteroids	Improvement
3	Itamura H (2020) [65]	NA	NA	NSCLC/III	RT 64 Gy	Pmebrolizumab	1.2	Shortness of breath on exertion, decreased spO2	Recurrent pneumonia	Stop ICIs/Corticosteroids	Improvement
4	Ye X (2022) [66]	M/64	Yes	NSCLC/IIIa	IMRT 50 Gy	Pembrolizumab	5	Cough and dyspnea	Patchy consolidation	Stop ICIs/corticosteroids	Improvement
5	Wang Y (2022) [68]	M/66	Yes	NSCLC/III	RT 60 Gy	Tislelizumab	4	Dyspnea, cough and fever	Interstitial pneumonia	Corticosteroids and antibiotics	Improvement
6	Torresan S (2023) [69]	M/77	Yes	NSCLC/IIIa	RT 54 Gy	Pembrolizumab	3	None	Pulmonary consolidation	Corticosteroids and antibiotics	Improvement
7	Xi L (2024) [70]	M/80	NA	SCLC/extensive	RT 40 Gy	Serplilimab	1.7	Fever and chest tightness	Interstitial fibrosis and inflammatory changes	Stop ICIs/Corticosteroids	Improvement
8	Zhu Y (2024) [71]	NA	No	NSCLC/IV	IMRT 60 Gy	Camrelizumab	5.5	Cough, dyspnea and fever	Interstitial pneumonia	Corticosteroids and antibiotics	Improvement
9	Schoenfeld J.D. (2019) [77]	M/64	NA	Melanoma/IV	RT 48 Gy	Nivolumab	2	Fatigue, shortness of breath, cough and diarrhea	Peripheral curvilinear consolidative opacities	Corticosteroids	Recurrent of pneumonitis

NA stands for “Not Available”.

## Data Availability

The original contributions presented in this study are included in the article/Appendix A. Further inquiries can be directed to the corresponding author.

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
