# Peer review of "A Systematic Review of Pneumonitis Following Treatment with Immune Checkpoint Inhibitors and Radiotherapy"

_biomedicines, 2025, doi:10.3390/biomedicines13040946_

Round 1
Reviewer 1 Report
Comments and Suggestions for Authors
Title: A Systematic Review of Pneumonitis Following Treatment with Immune Checkpoint Inhibitors and Radiotherapy
What is the main question addressed by the research?
The authors have tried to answer a research question “whether pneumonitis resulting from the combination of radiotherapy and immunotherapy treatments is more severe than when pneumonitis occurs with any one of the stated modes of therapies”?
- What parts do you consider original or relevant to the field? What
specific gap in the field does the paper address?
The paper addresses the concerns about number of events of pneumonitis associated with immunotherapy and or radiotherapy.
- What does it add to the subject area compared with other published
material?
It adds to the area of oncology and Pharmacotherapy / chemotherapy.
- What specific improvements should the authors consider regarding the
methodology? What further controls should be considered?
The authors shall address the comments in the manuscript reviewed attached / uploaded on the system.
In the abstract, some grammar and syntax errors are there.
Page 3 line 114-115, Write a table that best describes the search setting, then advanced setting with its relevant search phrases and BOOLEAN commands.
In page 15, Table 5, Why you presented Grade 3 to grade 5 together. Give a reason in the relevant section.
- Are the conclusions consistent with the evidence and arguments presented?
Were all the main questions posed addressed? By which specific experiments?
Conclusions need revision as it shall be based on the objective you set for your research questions. Though you have presented them in results section, yet here you shall be explicit to mention number of pneumonitis events in combination therapy or with the single therapy?
Conclusion seems to be a recommendation at this stage.
Your conclusions at the end of the manuscript and in the end of abstract shall also be the same. Revise it
- Are the references appropriate?
yes
. Any additional comments on the tables and figures and the quality of the
data.
Throughout the tables, the format of text is not same. The legends shall be in the landscape if the table is in landscape.

Author Response
Reviewer 1:
Comments and Suggestions for Authors
Title: A Systematic Review of Pneumonitis Following Treatment with Immune Checkpoint Inhibitors and Radiotherapy
1. What is the main question addressed by the research?
The authors have tried to answer a research question “whether pneumonitis resulting from the combination of radiotherapy and immunotherapy treatments is more severe than when pneumonitis occurs with any one of the stated modes of therapies”?
Reply: We clarified the scope of the systematic review.
2. What parts do you consider original or relevant to the field? What
specific gap in the field does the paper address?
The paper addresses the concerns about number of events of pneumonitis associated with immunotherapy and or radiotherapy.
Reply: Yes, in both the abstract and the main text, we have provided detailed explanations of the scope of the paper (Lines 15-17 and 122-124).
3. What does it add to the subject area compared with other published material?
It adds to the area of oncology and Pharmacotherapy / chemotherapy.
Reply: This was the primary objective of the systematic review.
4. What specific improvements should the authors consider regarding the methodology? What further controls should be considered?
The authors shall address the comments in the manuscript reviewed attached / uploaded on the system.
Reply: We included the assessment of the risk of bias using the Cochrane Risk of Bias Tool version 2 for randomized controlled trials and the RTI Risk of Bias Item Bank for non-randomized trials in the supplementary file.
In the abstract, some grammar and syntax errors are there.
Reply: We corrected the grammar and syntax errors in the abstract (Lines 11-39).
Page 3 line 114-115, Write a table that best describes the search setting, then advanced setting with its relevant search phrases and BOOLEAN commands.
Reply: We included two tables (Table 1 and Table 2 and Lines181-182) detailing the search settings and the databases.
In page 15, Table 5, Why you presented Grade 3 to grade 5 together. Give a reason in the relevant section.
Reply: Most patients experience pneumonitis as Grade 1–2 or Grade 3–5. For this reason, we present the data accordingly, with Grade 3+ considered clinically significant, as explained in the Results section (Lines 242-245 and 292-294).
5. Are the conclusions consistent with the evidence and arguments presented? Were all the main questions posed addressed? By which specific experiments?
Conclusions need revision as it shall be based on the objective you set for your research questions. Though you have presented them in results section, yet here you shall be explicit to mention number of pneumonitis events in combination therapy or with the single therapy?
Conclusion seems to be a recommendation at this stage.
Your conclusions at the end of the manuscript and in the end of abstract shall also be the same. Revise it.
Reply: We added the conclusion regarding the pneumonitis event in combination therapy to the conclusion section of the main text. Additionally, we revised both the conclusion in the text and the abstract to ensure they are consistent (Lines 30-39 and 501-510).
6. Are the references appropriate?
Yes
Reply: We thank the reviewer for his comment.
7. Any additional comments on the tables and figures and the quality of the data.
Throughout the tables, the format of text is not same. The legends shall be in the landscape if the table is in landscape.
Reply: We changed the legends to match the landscape orientation of the tables, which are also in landscape.
Reviewer 2 Report
Comments and Suggestions for Authors
The manuscript is well written and easy to follow. My only concern is with the tables—some are presented in a horizontal format, which makes them difficult to read. I suggest reformatting them to enhance readability and improve overall presentation. Additionally, if permitted by the journal’s policies, larger tables could be provided as separate Excel files rather than being merged into the main manuscript.
Author Response
Reviewer 2:
The manuscript is well written and easy to follow. My only concern is with the tables—some are presented in a horizontal format, which makes them difficult to read. I suggest reformatting them to enhance readability and improve overall presentation. Additionally, if permitted by the journal’s policies, larger tables could be provided as separate Excel files rather than being merged into the main manuscript.
Reply: We thank the reviewer for his comment. We changed the font of the table text to make it more readable. If the editor allows a separate Excel file for the tables, we will transfer them.

Reviewer 3 Report
Comments and Suggestions for Authors
The authors have carefully reviewed reports from randomized clinical trials, retrospective studies, and case reports to present the results in a tabular format that includes characteristics of the patients, disease, and treatment in cases of NSCLC. The advent of immunotherapy, in combination with radiotherapy and chemotherapy, offers the promise of improved cancer control, but with a cost of pneumonitis risk. By looking at reports from trials as well as "real-world" incidence, we may better learn how to predict, treat, and follow-up at-risk patients. This review presents data in a clear, comprehensive format. The discussion provides insight into the study limitations, comments on future study design and the importance of considering contributions of immunotherapeutics, as well as advanced radiation planning techniques in the development of severe adverse events, such as pneumonitis.
Section 3.4 requires minimal editing for typographical errors.
Although the authors included a comprehensive list of references, there are 2 published reviews of immunotherapy and radiotherapy in lung cancer that were not mentioned.
This review was published recently and may have come out after this paper was finished, but the authors may look at it to decide if it offers any other insights.:
CD73/adenosine dynamics in treatment-induced pneumonitis: balancing efficacy with risks of adverse events in combined radio-immunotherapies. Front Cell Dev Biol. 2025 Jan 13;12:1471072. doi: 10.3389/fcell.2024.1471072. eCollection 2024. PMID: 39872847 This review was published in 2018 and so may be outdated by now, but the authors should read it and decide if it holds anything of relevance. Combining Radiotherapy and Immunotherapy in Lung Cancer: Can We Expect Limitations Due to Altered Normal Tissue Toxicity? Int J Mol Sci. 2018 Dec 21;20(1):24. doi: 10.3390/ijms20010024. PMID: 30577587 Free PMC article. Review.Author Response
Reviewer 3:
The authors have carefully reviewed reports from randomized clinical trials, retrospective studies, and case reports to present the results in a tabular format that includes characteristics of the patients, disease, and treatment in cases of NSCLC. The advent of immunotherapy, in combination with radiotherapy and chemotherapy, offers the promise of improved cancer control, but with a cost of pneumonitis risk. By looking at reports from trials as well as "real-world" incidence, we may better learn how to predict, treat, and follow-up at-risk patients. This review presents data in a clear, comprehensive format. The discussion provides insight into the study limitations, comments on future study design and the importance of considering contributions of immunotherapeutics, as well as advanced radiation planning techniques in the development of severe adverse events, such as pneumonitis.
Reply: We appreciate the reviewer’s comments
Section 3.4 requires minimal editing for typographical errors.
Reply: We have corrected those errors in section 3.4 (Lines 303-326).
Although the authors included a comprehensive list of references, there are 2 published reviews of immunotherapy and radiotherapy in lung cancer that were not mentioned.
This review was published recently and may have come out after this paper was finished, but the authors may look at it to decide if it offers any other insights:
• CD73/adenosine dynamics in treatment-induced pneumonitis: balancing efficacy with risks of adverse events in combined radio-immunotherapies. Gockeln L, Wirsdörfer F, Jendrossek V. Front Cell Dev Biol. 2025 Jan 13;12:1471072. doi: 10.3389/fcell.2024.1471072. eCollection 2024. PMID: 39872847 This review was published in 2018 and so may be outdated by now, but the authors should read it and decide if it holds anything of relevance.
• Combining Radiotherapy and Immunotherapy in Lung Cancer: Can We Expect Limitations Due to Altered Normal Tissue Toxicity? Wirsdörfer F, de Leve S, Jendrossek V. Int J Mol Sci. 2018 Dec 21;20(1):24. doi: 10.3390/ijms20010024. PMID: 30577587 Free PMC article. Review.
Reply: We added those two reviews in the discussion section (Lines 389 and 398-406).
Reviewer 4 Report
Comments and Suggestions for Authors
In this study, Melina Yerolatsite and colleagues explore the risk of developing pneumonitis following the use of immune checkpoint inhibitors (ICI) in conjunction with radiation therapy (RT), a side effect of growing concern in cancer therapy. The researchers conducted a systematic review examining numerous studies, encompassing retrospective analyses, prospective clinical trials, and case reports, aiming to comprehend the frequency, intensity, identification, and management of pneumonia following this combined treatment approach. The study highlighted the difficulty in conducting a meta-analysis due to the high heterogeneity in the study methods and data reporting but observed that the incidence of pneumonia appeared to be higher in retrospective studies. In general, early detection and management of this type of pneumonia is critical to improving patient outcomes, and corticosteroids are the mainstay of treatment. Generally, the writing of this report fit the criteria, and the results supported the title. However, nearly 20 similar reports have been published earlier, and many of them included a meta-analysis, which weakens the significance of this report. The novelty of this report included the recruitment of different types of investigations, including retrospective studies, prospective clinical trials, and case reports. Second, the authors argued the treatment of corticosteroids in pneumonitis caused by ICI and RT, through the evidence from the case reports. The authors may need to emphasize the prediction and early treatment of pneumonia caused by ICI and RT with stronger evidence by investigating more related reports to increase scientific validity.
Author Response
Reviewer 4:
In this study, Melina Yerolatsite and colleagues explore the risk of developing pneumonitis following the use of immune checkpoint inhibitors (ICI) in conjunction with radiation therapy (RT), a side effect of growing concern in cancer therapy. The researchers conducted a systematic review examining numerous studies, encompassing retrospective analyses, prospective clinical trials, and case reports, aiming to comprehend the frequency, intensity, identification, and management of pneumonia following this combined treatment approach. The study highlighted the difficulty in conducting a meta-analysis due to the high heterogeneity in the study methods and data reporting but observed that the incidence of pneumonia appeared to be higher in retrospective studies. In general, early detection and management of this type of pneumonia is critical to improving patient outcomes, and corticosteroids are the mainstay of treatment. Generally, the writing of this report fit the criteria, and the results supported the title. However, nearly 20 similar reports have been published earlier, and many of them included a meta-analysis, which weakens the significance of this report. The novelty of this report included the recruitment of different types of investigations, including retrospective studies, prospective clinical trials, and case reports. Second, the authors argued the treatment of corticosteroids in pneumonitis caused by ICI and RT, through the evidence from the case reports. The authors may need to emphasize the prediction and early treatment of pneumonia caused by ICI and RT with stronger evidence by investigating more related reports to increase scientific validity.
Reply: In discussion section, we added the conclusion of another meta-analysis, which concludes that combination therapy provokes a higher incidence of pneumonitis. Additionally, we included articles explaining the effects of each method on the lungs. Finally, we also added the results of our systematic review to the discussion, in which most studies indicated that corticosteroids were the primary and most effective type of treatment (Lines 442-450 and 459-484).
Reviewer 5 Report
Comments and Suggestions for Authors
Overall good, but can be improved-
Abstract- no need to mention discussion instead write conclusion.
Include graphical abstract.
In introduction, write few sentences about Immune Checkpoint Inhibitors, may include animal study and mechanism of side effects.
Results
Present results in graph/table whereever possible.
All table must be modify to indicate references also include et al., where ever applicable.
Author Response
Reviewer 5:
Overall good, but can be improved-
Reply: We are grateful to the reviewer for their comment.
Abstract- no need to mention discussion instead write conclusion.
Reply: We made this change and wrote a conclusion instead of a discussion.
Include graphical abstract.
Reply: We included a graphical abstract.
In introduction, write few sentences about Immune Checkpoint Inhibitors, may include animal study and mechanism of side effects.
Reply: We added more details about the immune checkpoint inhibitors and the mechanisms of their side effects (Lines 50-57).
Results
Present results in graph/table wherever possible.
Reply: We have included all the results from the papers in tables.
All table must be modify to indicate references also include et al., where ever applicable.
Reply: We indicated the references from all included studies in the tables.
Round 2
Reviewer 1 Report
Comments and Suggestions for Authors
Nil